# Cellular and Molecular Processes in Wound Healing

**DOI:** 10.3390/biomedicines11092526

**Published:** 2023-09-13

**Authors:** Montserrat Fernández-Guarino, Maria Luisa Hernández-Bule, Stefano Bacci

**Affiliations:** 1Dermatology Service, Hospital Ramón y Cajal, Instituto Ramón y Cajal de Investigación Sanitaria (Irycis), 28034 Madrid, Spain; drafernandezguarino@gmail.com; 2Bioelectromagnetic Lab, Hospital Ramón y Cajal, Instituto Ramón y Cajal de Investigación Sanitaria (Irycis), 28034 Madrid, Spain; mluisa.hernandez@hrc.es; 3Research Unit of Histology and Embriology, Department of Biology, University of Florence, Viale Pieraccini 6, 50134 Firenze, Italy

**Keywords:** acute wounds, cellular infiltrate, chronic wounds, keloids, scars

## Abstract

This review summarizes the recent knowledge of the cellular and molecular processes that occur during wound healing. However, these biological mechanisms have yet to be defined in detail; this is demonstrated by the fact that alterations of events to pathological states, such as keloids, consisting of the excessive formation of scars, have consequences yet to be defined in detail. Attention is also dedicated to new therapies proposed for these kinds of pathologies. Awareness of these scientific problems is important for experts of various disciplines who are confronted with these kinds of presentations daily.

## 1. Introduction

The development of knowledge about wound healing (WH) throughout history has been remarkable. Historically, wounds were often related to infection, and patients consequently died from septicemia rather than from the wound itself. With the advent of light microscopy, a greater awareness of the fine cellular mechanisms of wounds has evolved, and this has aided our understanding of the treatment and care needed. Nevertheless, compromised WH is a major concern in public health. This is demonstrated by the fact that millions of people suffer from chronic wounds (CWs) [1,2].

## 2. Issues Related to the Study and Management of Wound Healing

Impaired WH appears to be a major concern in the public health sector, as expensive and complex treatments are necessary for its management. Millions of patients need care for chronic wounds and this costs thousands of millions in USD. This burden is increasing, mainly due to the growing incidence of aging, diabetes, and other risk factors in the population. A similar scenario is observed in Europe, where it is estimated that over 1.5 million people could be affected by CWs. As a result, WH is of great interest, and this is demonstrated by the number of fellowship programs offered for various medical professionals in this field, including vascular surgeons, nurses, dermatologists, and general practitioners. There is a demand for new technologies in WH to be introduced. Furthermore, additional problems exist because a comprehensive understanding of the biological mechanisms related to wound healing has not yet assumed popular relevance; rather, the focus is more directed toward niche problems, where, contrary to some disciplines, few scholars confront each other regarding the problems considered [1,2,3].

Guidelines undoubtedly represent an aid for wound care during clinical practice [4,5], in particular considering the fact that the biofilm, which maintains the CW in the inflammatory phase, is responsible for 80% of infections and must be removed [5]. Among the new techniques, it is particularly worth mentioning those concerning the effects of four basic compounds, i.e., octenidine (OCT), polyhexamethylene biguanide (PHMB), povidone–iodine (PVP-I), and sodium hypochlorite (NaOCl), as well as nanosilver, on cells and the inflammatory processes that act on the removal of the biofilm [5]. (See Section 8 for more information about recent therapies.)

## 3. Major Events in Wound Healing

Wound healing is divided into four overlapping phases: hemostasis, inflammation, proliferation, and remodeling/maturation [6,7].

**During hemostasis**, endothelial cells secrete the von Willebrand factor, favoring the adhesion of platelets, which release mediators. The release of these molecules results in a fibrin clot, which occludes the lesion and stops the bleeding. Due to the contraction of the smooth muscles (in response to the increased level of calcium ions), the vessels rapidly narrow, leading to the reduction of the flow. These events lead to the production of vasoactive metabolites that intervene in the vasodilation and relaxation of arterial vessels. The duration of this phase is a few minutes [8].

**During the inflammatory phase**, mast cells (MCs) promote vasodilation by the secretion of histamine or serotonin. This event involves the diapedesis of neutrophil granulocytes (Figure 1) and monocytes (cells that are ready to transform into macrophages). Consequently, phagocytosis is promoted inside the lesion against any pathogens or damaged cells. The leukocytes secrete cytokines and growth factors to initiate the proliferative phase. Other cell types such as keratinocytes participate in this event by releasing inflammatory cytokines. The duration of this phase averages 0–3 days [8,9]. It should also be noted that other molecules are involved in the inflammatory phase, such as cytokines, matrix proteins, or enzymes. Among these substances, chemokines are extremely important, mainly by attracting neutrophils and lymphocytes to coordinate the early stages of wound healing [10].

**During the proliferative phase**, fibroblasts, in addition to being involved in the formation of granulation tissue, are involved both in the regulation of keratinocyte migration and proliferation other than in angiogenesis. In addition to these cellular types, macrophages (Figure 1) and MCs continuously secrete growth factors involved in this process. The duration of this phase is 3–12 days [8].

The main processes that distinguish **the maturation (or remodeling) phase** are collagen restoration and wound contraction, the latter due to myofibroblasts (Smooth Muscle Actin (SMA)-positive) originating from the fibroblasts. The remodeling phase is also mediated by various growth factors that regulate the so-called transitions of the mesenchymal–mesenchymal and endothelial–mesenchymal phenotypes. In addition, these transformations occur through the Transforming Growth Factor (TGF) beta or the Notch signaling pathways, which inhibit the expression of cadherin in endothelial cells [11,12]. It should also be noted that Beta2AR has also been described as a critical molecule that mediates the epithelial–mesenchymal transition (EMT) process [11,12]. The duration of this phase ranges from 3 days to 6 months [8,9,13].

**The subsequent formation of the scar** involves a remodeling of the granulation tissue, where matrix metalloproteinases (MMPs) and their inhibitors (Tissue Inhibitor of Metalloproteinases (TIMPs) assume a crucial role. As a result, the formation of the extracellular matrix (ECM) is reduced, and its components are modified; in particular, type III collagen replaces type I collagen. The elastin absent in the granulation tissue appears again. It should also be considered that the cell death of some cell types in the granulation tissue undoubtedly constitutes a fundamental event in wound resolution. This paper also discusses whether the phenotype of the fibrocytes present in the dermis derives from myofibroblastic forms that progressively lose the typical morphological connotations or from other cellular forms that have differentiated late during the described processes [14] (Figure 1).

**Figure 1 biomedicines-11-02526-f001:**
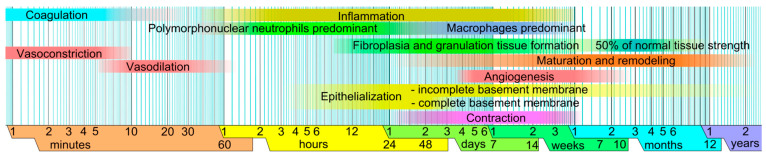
Times for the various stages of wound healing on a logarithmic scale, with faded intervals indicating significant variance, primarily based on the size of the lesion and the circumstances surrounding its healing [15].

## 4. Recent Advances in Understanding the Key Cells Involved in Wound Healing

### 4.1. Phase: Hemostasis

#### Platelets

Platelets are non-nucleated fragments of bone marrow megakaryocytes. As previously observed during the homeostasis phase, once activated, the platelets produce and secrete cytokines and growth factors such as Platelet-Derived Growth Factor (PDGF), TGFbeta, and Vascular Endothelial Growth Factor (VEGF), which play a role in clot development through various pathways (endocrine, paracrine, autocrine, or intracrine). Platelet activity (due to the secretion of various growth factors and cytokines) is also involved in various phases that characterize wound healing. In fact, they promote (i) the migration, proliferation, and stimulation of fibroblasts; and (ii) the differentiation of endothelial cells in new blood vessels and of the mesenchymal stem cells in specific cellular types. Other growth factors, such as PDGF, are not only involved in the activation of neutrophils and macrophages, but they also act as a mitogenic and chemoattractant for fibroblasts and smooth muscle cells [16]. In addition, in response to the cell damage evoked by the lesion, platelets stimulate the inflammatory response via the secretion of prostaglandins, leukotrienes, and thromboxanes [17] (see Table 1 for other details).

### 4.2. Phase: Inflammatory

#### 4.2.1. Mast Cells

The number of MCs (Figure 2B) and their degranulation index increases at the edge of a lesion within 3 h, and these values become lower than baseline values after 6 h or more. In the following 10 days, the MCs increase again in number and then decrease to the control values 21 days after the production of the skin lesion. The increase in these cellular types is probably related to the increase in expression of Monocyte Chemoattractant Protein (MCP) 1 and the subsequent secretion of TGFbeta, a chemoattractant for MCs. Through the release of Tumor Necrosis Factor (TNF) alpha, these cells contribute to hemostasis by increasing the expression of Factor XIIIa in dermal dendrocytes, thus contributing to the stabilization of the clot. MCs also release inflammatory mediators (histamine, VEGF, IL6, IL8, and TNFalpha) that contribute to endothelial permeability, thus facilitating the migration of monocytes and neutrophils toward the lesion. The release of leukotrienes, proteases, and cytokines by MCs is related to the production of chemotactic signals for the various types of granulocytes, while the release of tryptase and cathepsin G plays an important role in the endothelial–leukocyte interactions. If angiogenesis is linked to the secretion of TGFbeta, then VEGF, chymase, tryptase, and heparin (secreted by MCs) can inhibit angiogenesis by interacting with inhibiting pro-angiogenic factors. Finally, it should be noted that during the maturation and remodeling processes, the MCs (i) activate fibroblasts, promoting collagen synthesis (this effect is partly mediated by tryptase); (ii) secrete growth factors and cytokines which regulate the transition between fibroblasts and myofibroblasts; and (iii) release matrix metalloproteinases, thus initiating the degradation of the extracellular matrix I [18] (see Table 1 for more details).

#### 4.2.2. Neutrophils

Neutrophils (Figure 2A) reach the wound in the hour following the insult and infiltrate the injured site in the following hours. The migration of these cellular types to the wound area is favored by PDGF or Connective Tissue Chemokine Activating Peptide (CTAPIII), released by platelets, which are proteolytically activated by Neutrophil-Activating Peptide 2 (NAP2) and CXCL7 in neutrophils [19]. On the other hand, low concentrations of NAP2 promote the chemoattraction of neutrophils via CXCR2; Growth Related Oncogene (GRO) alpha secretion by endothelial cells and pericytes further promotes the neutrophil migration process [20]. Neutrophil recruitment is also supported by ENA78 (CXCL5), which is expressed at lower levels than GROalpha in mononuclear cells within the matrix [21]. The infiltration of these cells is related to (i) signaling and chemotactic molecules, such as damage-associated molecular patterns (DAMPs) released by necrotic cells; and (ii) TGFbeta, C3a, C5a, and hydrogen peroxide, released by platelets. DAMPs operate directly on neutrophils by attaching to particular Pattern Recognition Receptors (PRRs), such as the Toll-like receptor (TLR) or by promoting the production of chemoattractants in other cells. CXCL1, CXCL2, and CXCL8 also play a significant role in the recruitment of inflammatory cells. The pro-inflammatory mediators TNFalpha, IL1beta, IL6, and CXCL8 are also released by neutrophils once they have aggregated at the site of the lesion, thereby further promoting inflammation. Neutrophils promote wound infections by clearing the wound of any pathogens and generating a heavily oxidized environment via the production of reactive oxygen species (ROS). Additionally, they secrete antimicrobial proteins as well as “traps” of chromatin and protease that kill bacteria in the surrounding microenvironment. Neutrophils aid angiogenesis and the growth of fibroblasts and keratinocytes by boosting the release of cytokines (IL1beta and IL8), growth factors (VEGF), and chemokines such as MCP1. The disappearance of neutrophils from the site of the lesion occurs after 24–36 h by apoptosis when the circulating monocytes (attracted by the IL8 secreted by the neutrophils themselves) enter the wound and differentiate into macrophages (see Table 1 for more details).

#### 4.2.3. Macrophages

Macrophages reach the lesion area after 48 and 72 h, attracted by the release of chemical messengers released by various cellular types. Within their granules, macrophages contain growth factors, such as TGFbeta and Epidermal Growth Factor (EGF), which, in turn, are involved in the inflammatory response, in angiogenesis, and in the formation of granulation tissue. Macrophage activation leads to the formation of two cell subtypes called M1 and M2. The formation of the M1 phenotype, mediated by TLR44 and Interferon (IFN) gamma ligands, results in a pro-inflammatory macrophage. The appearance of the M2 phenotype (mediated by IL4 and/or IL13) involves the release of some growth factors, such as PDGF, Fibroblast Growth Factor (FGF) alpha, FGFbeta, TGFalpha, and TGFbeta. FGFbeta is a chemotactic and mitogenic factor for fibroblasts, endothelial cells, and other mesenchymal cells, which are, in turn, involved in angiogenesis [15]. But research on humans has revealed that different macrophage subtypes are based on changes in the macrophage phenotype during the healing of wounds, which have been defined based on their cell surface markers and the production of cytokines, growth factor, and chemokines. During the proliferative phase, macrophages play a particularly important role in angiogenesis, as they position themselves around the newly formed blood vessels to help their subsequent differentiation. In fact, IL-1 produced by macrophages stimulates the proliferation of endothelial cells and promotes angiogenesis. Furthermore, FGFbeta and other growth factors and cytokines (TNF, IL6, and IL1) [16] are involved in the formation of collagen, fibronectin, and proteoglycans, as well as processes associated to wound contraction and epithelialization, while TNFalpha is a mitogen for fibroblasts. Macrophages, together with keratinocytes and endothelial cells, produce MCP1, which attracts monocytes, MCs, and lymphocytes during skin wound healing. Furthermore, MCP1 may also contribute to endothelial cell migration during angiogenesis [22]. During the remodeling phase, macrophages release MMPs to reduce the temporary extracellular matrix and undergo apoptosis [7,23] (see Table 1 for more details).

#### 4.2.4. Dendritic Cells

Cellular interactions between dendritic cells (DCs) and MCs are necessary for the secretion of soluble factors such as TGFbeta or TNFalpha, which are crucial for the differentiation of fresh precursors into DCs. DCs are rarely seen in wounds less than 1 day old, whereas the accumulation of DCs is seen in wounds aged 3 to 14 days. Within the dendritic cell population there is a cell subpopulation present in the human epidermis expressing CD1a and CD207/langerin, which are the so-called Langerhans cells. Studies have shown that these cell types rapidly increase in number within the first hour after injury. In confirmation of these hypotheses, it was verified that in a mouse model, complete healing of the wound did not occur in cases of DC deficiency. Consequently, they intervene in the initial phase of wound healing, secreting factors that trigger the proliferation of cells involved in this phenomenon [24,25] (see Table 1 for more details).

#### 4.2.5. Plasmacytoid Dendritic Cells

Plasmacytoid dendritic cells (PDCs) (Figure 2C) produce large amounts of type I IFN and express TLR7 and TLR9. Although absent from normal skin, PDCs are rapidly recruited upon injury; through interaction with Treg cells [26], they secrete type I IFN, which contributes to wound healing (see Table 1 for more details).

#### 4.2.6. Lymphocytes

These reach the site of the lesion 72 h after the injury. These cells are attracted to the wound by IL1, of macrophage origin, which is also involved in the remodeling of the extracellular matrix. The lymphocyte response is evoked by released factors such as IFNgamma and TNFalpha present in the skin microenvironment, as well as by antigen-presenting cells. Once inside the lesion site, lymphocytes also produce various chemokines; initially, MCP1, and after day 4, IP10 (CXCL10), MIG (CXCL9), and MDC (CCL22) [7,8]. These cells also secrete growth factors, such as EGF and FGFbeta. Furthermore, recent studies indicate that Tregs facilitate healing in skin wounds by inducing the expression of the EGF receptor. Removal of EGFR in Tregs results in the non-activation and subsequent non-migration of these cells, the accumulation of proinflammatory macrophages, and delayed wound closure [27] (see Table 1 for more details).

### 4.3. Phase: Proliferation

#### 4.3.1. Keratinocytes

Keratinocytes (Figure 3 and Figure 4) are directly involved in the wound-healing process. These cells become active during the inflammatory phase of wound healing, secreting various cytokines and growth factors.

Due to the morpho-functional changes, the keratinocytes migrate toward the injured area in a period of approximately 24 h. The process, characterized by detachment from the underlying basement membrane, is facilitated by MMP1, which is expressed at high levels at the wound edges. Migrating basal keratinocytes express CD44, in contrast to resting basal cells [28,29,30]. The migration and proliferation of keratinocytes is stimulated by IL8 and is highly expressed at the wound margin to promote wound closure [31,32]. Furthermore, during the migratory process, keratinocytes express CXCR2, the IL8 receptor, and GROalpha [33]. Interaction with a provisional matrix of fibronectin cross-linked fibrin promotes the adhesion of keratinocytes to the matrix at the base of the wound. However, the mechanisms of re-epithelialization are not fully understood. During wound healing, keratinocytes modulate the activity of fibroblasts by stimulating their proliferation and the secretion of growth factors, which are, in turn, involved in the proliferation of these types of cells. By recognizing Pathogen-Associated Molecular Patterns (PAMPs) and DAMPs, the keratinocytes determine the activation of pro-inflammatory signaling pathways through the production of interleukins (e.g., IL1, IL6, IL8, IL10, IL18, or IL20), TNFalpha, and chemokines (e.g., Regulated Upon Activation Normal T-cell Expressed and Secreted (RANTES), MCP1, or Macrophage Inflammatory Protein (MIP1)). Finally, the differentiation of fibroblasts into myofibroblasts can be modulated by keratinocytes (Figure 3) in relation to a finely tuned balance between a pro-inflammatory or TGFbeta-dominated environment [28,29,30] (see Table 1 for more details).

#### 4.3.2. Endothelial Cells

Endothelial cells act during the proliferative phase of wound healing. Indeed, these cells mediate the recruitment of leukocytes from the blood to the tissues, and they are protagonists of angiogenesis, which are mediated by, among others, substances such as MCP1, RANTES, IL8, GROalpha, IFNgamma-inducible Protein (IP10), and Monokine-induced Gamma Interferon (MIG) [34]. Furthermore, the Glu-Leu-Arg (ELR) motif [35] close to the cysteine at the NH2-terminus of various chemokines has been shown to be a potent promoter of angiogenesis. This group of chemokines containing the ELR motif includes GROalpha, GRObeta (CXCL2), and GROgamma (CXCL3), as well as CTAPIII, beta-thromboglobulin, and NAP2 [36]. Angiogenesis is inhibited by CXC chemokines lacking the ELR motif, such as (IP10), MIG, or Interferon-inducible T-cell Alpha Chemoattractant (ITAC) [35,36]. MCP1 expression in these cells creates a chemokine gradient that aids in the attraction of particular leukocytes to inflamed areas [37]. As regards the passage of leukocytes toward the injured area, the variations in permeability of the same blood vessel are linked to the activity of VEGF. Other growth factors (FGF, PDGF, and Stromal-derived Growth Factor SDF1), however, modulate the proliferation and chemotaxis of endothelial cells. It should also be noted that the activity of VEGF and FGF is, in turn, regulated by proteoglycans and by the syndecans of heparin sulfate. The migration and proliferation of endothelial cells is a complex process involving changes in the reorganization of the cytoskeleton and changes in adhesion molecules and in signal transduction. The stabilization of angiogenesis is, however, linked to the recruitment of pericytes [38,39]. It should also be noted that any repair processes of the endothelial cells during angiogenesis are related to the so-called endothelial progenitor cells (EPC), which exhibit markers for endothelial cells and hematopoietic stem cells, including CD146, von Willebrand Factor (VWF), and VEGFR2, and are a source of pro-angiogenic cytokines [16] (see Table 1 for more details).

#### 4.3.3. Pericytes

Pericytes (Figure 2D) are a new cell type involved in the wound-healing phenomena. These cells, together with endothelial cells, participate in neo-angiogenesis as well as contributing to the caliber of the newly formed vessel. Studies have shown that pericytes can act in synergy with macrophages in the inflammatory phase of wound healing. It also appears that these cells stimulate an immune response by triggering the activation of T lymphocytes through the secretion of various cytokines. Recent studies have demonstrated how pericytes can differentiate into myofibroblasts via a PDGF-mediated mechanism [40] (see Table 1 for more details).

#### 4.3.4. Fibroblasts

Fibroblasts (Figure 4) play a critical role during the various stages of wound healing by secreting extracellular matrix during the proliferative stage and in subsequent remodeling. Early phases of repair are mediated by fibroblasts located in the reticular dermis, while those located in the papillary dermis are often recruited during the re-epithelialization phase. During the inflammatory phase, fibroblasts stimulate the local immune response in various ways: (1) secreting pro-inflammatory cytokines and growth factors (TNFalpha, IFNgamma, IL6, and IL12) and releasing a wide range of chemokines (CXCL1, CX3CL1, and CCL2); (2) promoting cellular interactions through the expression of ICAM1 (CD54) and CD40, which stimulate dendritic cell activity; (3) secreting MMPs necessary for stromal remodeling of a wound. Cellular interactions between fibroblasts and macrophages regulate the passage from the inflammatory to the proliferative phase, thus accelerating the healing of wounds. Fibroblasts subsequently migrate and proliferate due to the secretion PDGF, TGFbeta, FGF, VEGF, and other growth factors. During the proliferative activity, the fibroblasts, via the release of chemicals as VEGF, FGF, Angiopoietin (ANG) 1, or Thrombospondin (TSP), contribute to vascularization. Recent studies suggest that, at this stage, fibroblasts are targets of chemokines, such as MCP1, that increase the manifestation of MMP1 and TIMP1 in these cells [22]. ECM synthesis by fibroblasts supports re-epithelialization by also acting as a support for myofibroblasts during their function [41] (see Table 1 for more details).

### 4.4. Phases: Maturation and Remodeling

#### 4.4.1. Myofibroblasts

The contraction of the granulation tissue formed following the injury is promoted by myofibroblastic cells, which differentiate from fibroblasts in situ. Myofibroblasts regulate wound contraction and tissue remodeling by assuming a contractile phenotype (due to the presence of actin) and releasing extracellular matrix molecules. The differentiation of fibroblasts into myofibroblasts seems to be regulated by the presence of TFGbeta1 in the skin microenvironment and probably by the same conditions of non-deformability of the ECM regulated by molecular modifications (the replacement of collagen III with collagen I is one example), which confer tensile strength [41,42]. Subsequently, after having restored the integrity of the tissue, the activities of the myofibroblasts cease, and a part of their population is eliminated due to the apoptotic phenomena; however, the modulation of the various signals (including timing) that trigger these phenomena is not fully understood [41,42] (see Table 1 for more details).

#### 4.4.2. The Central Role of Mast Cells in Wound Healing: A Hypothesis

Reflecting on the salient data of this review, it seems appropriate to suggest a central role for MCs during the healing of wounds. In acute wounds, MC activation, by various cell types or by the microenvironmental stimuli themselves, leads to the release of TNFalpha for dendritic cell differentiation, the secretion of various substances capable of inducing angiogenesis, the release of extracellular matrix by the fibroblasts, and, therefore, to the response of the cellular infiltrate in response to the injury. These events are also related to the synthesis of TGFbeta by many cells present in the microenvironment and produce, as the main result, the differentiation of macrophages into the M1 and M2 phenotypes. The latter type of cells, probably together with keratinocytes, activate fibroblasts that differentiate into myofibroblasts. TGFbeta activity also appears to be related to the formation of PDCs, which can interact with Treg cells to induce tolerance through the co-expression of CD45 by these latter cell types during wound-healing processes. In chronic wounds, there is further evidence for the important role of these cells. In various articles, it has been reported that mast cells show an increased degranulation index in chronic wounds. This event leads to modifications of the inflammatory infiltrate, with various responses from the various cell types involved. Among the cytokines produced here is TGFbeta, which plays a crucial role in the various stages leading to the healing of chronic wounds. In this type of wound, as already noted, an interaction between MCs and neuronal cells is described. The release of the nerve mediators involved in wound healing is, in fact, linked to the interaction between these types of cells [18].

**Table 1 biomedicines-11-02526-t001:** The main bioactive factors secreted by cells involved in wound healing.

Wound Healing				
**Phases**	**Time**	**Cells**	**Main Bioactive Factors Secreted by Cells Involved in Wound Healing**	**Functions**
**Hemostasis**	**A few minutes**	**Platelets**	**CYTOKINES:** TNFalpha **GROWTH FACTORS:** PDGF, TGFbeta, TGFalpha, FGF, IGF1, VEGF **CHEMOKINES:** CXCL8, CXCL1, CXCL2	Initiation of inflammatory responses, angiogenesis
**Inflammatory**	**3–12 min to 3 days**	**Mast Cells**	**BIOGENIC AMMINES:** Histamine **CYTOKINES:** TNFalpha, IL4, IL6, IL8 **GROWTH FACTORS:** VEGF, FGF	Vasodilation, inflammatory response, production of ECM
**Inflammatory**	**3–12 min to 3 days**	**Neutrophils**	**CYTOKINES:** IL1beta, IL6, IL8, TNFalpha **CHEMOKINES:** CXCL1, CXCL2, CXCL8 **GROWTH FACTORS:** IGF, VEGF	Inflammatory response, keratinocyte proliferation, fibroblast proliferation, angiogenesis, collagen synthesis, endothelial cell activation
**Inflammatory**	**3–12 min to 3 days**	**Macrophages**	**CYTOKINES:** IFNgamma, IL1beta, IL6, IL8, IL10, TNFalpha **CHEMOKINES:** RANTES **GROWTH FACTORS:** EGF, FGF, IGF, PGDF, TGFbeta, VEGF	Inflammatory response fibroblast proliferation, fibroblast chemotaxis, angiogenesis, ECM deposition
**Inflammatory**	**3–12 min to 3 days**	**Dendritic cells,** **plasmacytoid dendritic cells**	**GROWTH FACTORS:** TGFbeta **CYTOKINES:** IFN gamma	Inflammatory response
**Inflammatory**	**3–12 min to 3 days**	**Lymphocytes**	**CYTOKINES:** IFNgamma, IL2, IL4, IL10 **CHEMOKINES:** MCP, RANTES, MIP, Lymphotactin	Inflammatory response, decrease in collagen synthesis, synthesis of MMPs
**Proliferation**	**3 days to 12 days**	**Keratinocytes**	**CYTOKINES:** IL1, IL6, IL8, IL10, IL18, IL20, TNFalpha **GROWTH FACTORS:** TGFbeta, VEGF, EGF, PGDF, **CHEMOKINES:** RANTES, MCP or MIP-1	Proliferation of keratinocytes, angiogenesis, proliferation of keratinocytes, inflammatory response
**Proliferation**	**3 days to 12 days**	**Endothelial cells**	**GROWTH FACTORS:** FGF, IGF, TGFbeta, PGDF, VEGF	Proliferation of fibroblasts and keratinocytes, differentiation of keratinocytes, angiogenesis
**Maturation or remodeling**	**12 days to 6 months**	**Fibroblasts**	**CHEMOKINES:** CXCL1, CX3CL1, CCL2 **CYTOKINES:** IL6, IL8, IL12 **GROWTH FACTORS:** FGF, IGF, KGF, VEGF	Chemotaxis of inflammatory cells, proliferation of fibroblasts, fibroblast differentiation

Adapted from [7,43].

## 5. Molecular Events in Wound Healing

### 5.1. Growth Factors, Cytokines, and Other Substances

Wound healing is a complex process essentially controlled by molecules such as cytokines and growth factors, which are released at different stages of wound healing. The modulation of this process is extremely important since any variation of it involves the possible lack of healing of the wound and, with it, the creation of conditions for the formation of a chronic wound. In particular, the pro-inflammatory cytokines, IL1beta, TNFalpha, and IL6 serve to attract inflammatory cells to the wound site. Within the site of the lesion, the inflammatory cells release various growth factors, such as PDFG and TGFbeta, which draw proliferating fibroblasts to this area. It should also be noted that macrophages and active fibroblasts release FGF2 (bFGF), Keratinocyte Growth Factor (KGF), FGF7, EGF, Hepatocyte Growth Factor (HGF), TGFalpha, and Insulin-like Growth Factor (IGF) 1 to stimulate epithelialization. VEGF and PDGF, produced by fibroblasts, keratinocytes, and macrophages, activate the endothelial cells to initiate the angiogenesis process.

Other factors also participate in this process: examples are transcription factors (the E2F family) and signaling pathways (Wnt/beta catenin), or other molecules, such as the Signal Transducer and Activator of Transcription (STAT) 3, homeobox genes, hormone receptors (androgens, estrogens, and glucocorticoids), Peroxisome Proliferator-activated Receptors (PPARs), Activator Protein (AP) 1, c-Myc, and *ETS-related Gene (Erg)1*, proteases (including MMPs), cytoskeleton proteins, and enzymes that regulate the cellular redox balance. These elements are interconnected rather than independent (see Table 1 for details about the secretion of bioactive factors during wound healing) [7,43].

Invertebrates and vertebrates both exhibit distinct transcription-independent diffusible signals during wound healing. Without a doubt, hydrogen peroxide (H_2_O_2_) and adenosine (for the further autocrine release of ATP) play crucial roles.

In general, Protein Kinase C (PKC), Ca^2+^/Calmodulin-dependent Protein Kinase (CaMK) 4, and ROS modify genetic transcription as a result of the lesion’s rapid increase in intracellular Ca^2+^ concentration, which is involved in cell communication, migration, adhesion, inflammatory responses, angiogenesis, and re-epithelialization. It should also be noted that tissue damage stimulates Ca^2+^ waves that activate the RHO family GTPases, boosting actin polymerization and actomyosin contractility to preserve stromal integrity. Furthermore, the release of Ca^2+^ enhances various signaling pathways (that of c-Jun N-Terminal Kinase (JNK) and Mitogen-Activated Protein Kinase (MAPK), which induce both the activation of the transcription factors and the increase in insult-response genes, such as those of the cytoskeleton. The release of ATP and its activation by purinergic receptors influence the wound-healing process. During the insult, DNA damage in epithelial cells is recognized by P2Y receptors on nearby healthy cells; these are capable of transmitting cytoplasmic signals involving the activation of intracellular Ca^2+^ and MMPs. This mechanism ensures the release of specific growth factors (e.g., EGF) capable of activating the various cascade mechanisms involved in wound healing [44].

### 5.2. Genetic Activation in Wound Healing

The different stages of wound healing and their overlap are characterized by the functionality of genes coding for various molecules (cytokines, chemokines, and growth factors). More specifically, in the initial stages of wound healing, one of the highly differentially expressed genes (DEGs) and one hundred of the less manifested DEGs were identified. *Tyrosinase (TYR)*, *Tyrosinase-Related Protein 1 (TYRP1)*, and *Dopachrome Tautomerase (DCT)* are hub genes that play a role in the development of melanin. A total of 85 DEGs and 164 downregulated proteins were found to be present during the inflammatory and proliferative phases. *Checkpoint Kinase 1 (CHEK1)*, *Cyclin B1 (CCNB1)*, and *Cyclin-dependent Kinase 1 (CDK1)* genes are hub genes involved in the cell cycle and the P53 signaling pathway. A total of 121 DEGS and 49 weakly expressed genes were identified throughout the remodeling phase. The genes for *Collagen Alpha Chain 1 (COL4A1)*, *Collagen Type 4 Alpha Chain 2 (COL4A2)*, and *Collagen Type 6 Alpha Chain 1 (COL6A1)* are hub genes involved in the digestion and absorption of proteins and in the interaction with the ECM receptor [45]. It should also be noted that, in recent decades, scientific attention has been directed toward the effects each cytokine has on distinct parameters of WH in several experimental scenarios. Recent studies have shown that the key genes involved are *IL1beta*, *IL6*, *CCL4*, *CXCL1*, *CXCL2*, *CXCL3*, *CXCL5*, *CXCL6*, *and CXCL10*, which play essential roles in the interactions between various receptors and the IL17 signaling pathways. It has also been established that IL6 and IL1beta are essential for skin healing and for increasing keratinocyte motility. Finally, it should be noted that further studies have demonstrated that the low expression of CXCL1 and CXCL5, both chemoattractants for neutrophils, inhibits mouse WH [45].

Pro-inflammatory genes encoding the activation of molecules such as TNFalpha, IFNgamma, or TGFbeta are expressed very early after injury. As wound healing progresses, the gene profile includes genes coding for molecules such as *VEGF*, *PDGF*, *FGF2*, *and MMP*, which promote the proliferation of fibroblasts and keratinocytes, as well as epithelialization, angiogenesis, and the initiation of eventual repair. During the remodeling phase, the genes encoding *TGFbeta1 and MMP* expression are upregulated to promote collagen production by fibroblasts and the removal of ECM in tissue resorption. Any alteration in gene expression, and therefore in the release of factors (cytokines, chemokines, and growth factors), can affect the healing sequence, which can lead to the development of chronic wounds [7].

**Epigenetic mechanisms** are involved in wound healing, although the knowledge of the molecular mechanism is limited. To date, various examples have accumulated regarding these mechanisms, including the discovery of various microRNAs, which are respectively engaged in the regulation of inflammatory responses, in the production of the ECM, in the activation of cell proliferation, and in the communication between cells during the processes related to wound healing (for more details see ref [44]).

An example **of the post-translational mechanisms** is provided by the proteolytic lysis of fibronectin that induces cell proliferation and migration during wound healing (for more details see ref [44]).

It has been observed that the wound-healing process is characterized by **a controlled quality trait,** which differs between individuals. In fact, in some studies, it has been demonstrated that some lines of mice (e.g., MRL/MpJ-Faslpr (MRLF)) are able to heal an ear-punched hole (2 mm diameter) in 30 days, while in others (C57BL/6 or SJLJ), there is a healing equal to 40 and 25%, respectively, over the same duration. Other studies have shown that in MLRF mice, there is the presence of a genetically controlled trait. Therefore, the MRLF mouse can be considered a model for the evaluation of the molecular mechanisms of wound repair in mammals [44].

The above paragraph, dedicated to genetic activation, shows that the research on this is still ongoing and, therefore, the data are often confused and even contradictory. However, it must be said that this research is progressing, and one should not be surprised if today’s studies become the basis for tomorrow’s therapies (for more details see refs. [44,45]).

## 6. Malfunctioning of Processes Related to Wound Healing

### 6.1. Chronic Wounds

If the wound-healing processes are not completed within 6–8 weeks, the wound is considered chronic, and its treatment is extremely expensive [1,46,47]. Although there are different types of chronic wounds, a common mechanism is the formation of the biofilm, which is one of the main causes of delay in wound healing, thereby promoting the evolution of a wound from acute to chronic. Wound microbial communities are characterized by the presence of various species of bacteria occupying the site of infection.

Aerobes and facultative bacteria such as *Staphylococcus aureus*, *Pseudomonas aeruginosa*, and β-hemolytic *Streptococci* are the leading origin of CWs, and they make up a significant proportion of the microbial population; in addition, there are *Enterococcus* species, *Klebsiella pneumoniae*, *Acinetobacter baumannii*, and *Enterobacter* species (ESKAPE pathogens), coagulase-negative *Staphylococci*, and *Proteus* species.

Many endogenous fungi, including *Candida*, *Curvularia*, and *Malasezzia*, have been associated with CW infections [48].

In general, changes in MMP secretions perpetuate the inflammatory stage in CWs compared to acute wounds. Furthermore, an excessive inflammatory response in CWs is mediated by many cells belonging to the cellular infiltrate [46,47,49,50]. Neutrophils appear in excess in CWs and release significant loads of metalloproteinases, which not only destroy the connective tissue matrix and elastase but also inactivate important factors involved in wound healing, such as PDGF and TGFbeta. The cellular interactions of keratinocytes with immune cells of the cellular infiltrate via the secretion of various signaling molecules need to be considered, but the contribution of such mechanisms to the formation of a CW is not fully understood. Furthermore, in chronic wounds, keratinocytes express genes involved in partial proliferative activation, and this explains the increased epidermal proliferation that occurs at the ulcer edges. In addition to this, fibroblasts do not respond to the TGFbeta stimulus regarding their migration. Indeed, reduced levels of TGFbetaR, and that of the downstream components of the TGFbetaR-signaling cascade, have been noted [46,47,49,50]. Finally, it should be remembered that neuroimmunomodulation can take on an important significance in the modulation of the cicatricial processes of chronic wounds. Cellular interactions have recently been observed between MCs and neurons containing mediators implicated in processes related to wound healing, such as calcitonin-gene-related peptide (CGRP), nerve growth factor (NGF), neurokinin A (NKA), neuropeptide Y (NPY), substance P (SP), protein gene product (PGP) 9.5, and vasoactive intestinal peptide VIP [49]. It is assumed that the cellular interaction between cells of the immune system and neuronal cells may explain some of the previously described phenomena, such as the excessive secretion of ECM by fibroblasts, as well as the increased TGFbeta levels and the response of cellular infiltrates [49].

### 6.2. Wound Healing in Conditions of Hyperglycaemia: Chronic Diabetic Wounds

Fifteen percent of patients with Type 2 Diabetes Mellitus (T2DM) develop diabetic foot ulcers (DFUs), often leading to amputation or death [51]. Therefore, there is a critical need to clarify the pathogenic mechanisms that underlie the ulceration process and that affect wound healing in diabetics. Diabetic patients typically experience wounds with reduced epithelialization and prolonged inflammation, which impairs healing. Regarding the inflammatory phase, the increase in neutrophils and the consequent increase in the ratio of neutrophils to lymphocytes is considered a hallmark of the impaired wound healing observed in individuals with T2DM. However, it is a common opinion that the cause of many alterations of the inflammatory response is not related to the high glucose levels recorded but to epigenetic variations of the immune cells before their entry into the wound due to related systemic complications.

During the systemic inflammation observed in diabetic patients, there is an inhibited migration of Treg lymphocytes and an increased infiltration of Th17 cells into the diabetic wound. These events are undoubtedly associated with excessive neutrophilic inflammation and, consequently, with a prolonged inflammatory phase.

During the proliferative phase within DFUs, skin cell proliferation and the activation of stem and progenitor cells are significantly reduced. Although the event described can be considered linked to the effect of the prolonged inflammatory phase, other factors act in this context, such as the protein glycation mediated by T2DM, the altered angiogenic functionality, and the consequent oxidative stress. It should also be taken into account that in diabetic wounds, the polarization of monocytes toward M2 macrophages is interdicted and pro-inflammatory, i.e., M1 macrophage polarization is stimulated.

Finally, during wound remodeling, T2DM-compromised fibroblasts show signs of inactivation, resulting in decreased collagen deposition and signaling, including downregulation of the TGFbeta pathway.

Thus, fibroblasts in damaged wounds have their ECM deposition capacity significantly reduced [51].

## 7. Fibrosis: Hypertrophic Scarring and Keloids Associated with Wound-Healing Phases

In both pathologies, i.e., hypertrophic scarring and keloids, it is noted that myofibroblasts are not replaced by fibroblasts, which, instead, are stationed in the proliferative phase of the wound with limited signs of apoptosis [52]. Myofibroblasts are therefore capable of generating persistent chronic inflammation due to the secretion of various cytokines, chemokines, and growth factors (TGFbeta1, TGFbeta2, VEGF, FGF, and Connective Tissue Growth Factor (CTGF)). Furthermore, these cells are able to synthesize large quantities of extracellular fibrotic matrix, consisting of collagen I, III, and alphaSMA, which alter the normal function of the various tissues [52].

### 7.1. Hypertrophic Scarring

Hypertrophic scarring is a serious clinical problem, affecting millions of patients in advanced countries. A hypertrophic scar shows fibrous tissue limited to the scar, unlike a keloid, which exceeds the limit of the initial lesion. An excessive activation of fibroblasts, originating from the generation of profibrotic molecules that are, in turn, supported by the activation of an exaggerated inflammatory response or by the increase in angiogenesis, seems to be the origin of a hypertrophic scar. It should also be noted that, in recent years, numerous biomolecules implicated in this pathology have been discovered, but their meaning has not been clarified [53,54,55].

The main treatment for hypertrophic scars is prevention where possible, especially in a planned surgery. When therapies need to be added, the option is usually a combination, depending on the availability or patient preference, of intralesional corticoids, pressure therapies, patching with different substances, or laser treatment, such as with fractionated CO_2_ or erbium lasers, are probably the most widely used [56,57].

### 7.2. Keloids

Keloids are fibrotic skin lesions that result from a variation of the normal wound-healing process of the skin. The growth of a keloid can be triggered by any type of skin lesion. Keloid scars originate from an excessive proliferation of fibroblasts with a consequent reduction in the apoptotic rate and an increase in angiogenesis, with a consequent increased infiltration of inflammatory cells. Recently, Aghmandy et al. proposed that normal signaling in keratinocytes and fibroblasts is impaired and that the keloid methylation status is implicated in the origin of this pathology [58]. In addition, some mutations have been described in the Asian population, suggesting an individual predisposition, as seen in habitual clinical practice [53].

The treatment of keloids is challenging as they tend to be recurrent and resist therapies, which habitually are used in combination due to this resistance. There are several treatments available: intralesional corticoids, silicone patches, bleomycin, cryotherapy, radiotherapy, fractionated laser treatment, and blue LED light irradiation [4,55,56,57,59].

## 8. Impairment of Wound Healing and Recent Therapeutic Strategies

Wound care represents an important problem in medicine today. This section of this review is dedicated to the most innovative therapies proposed for this issue.

The presence of epidermal stem cells (ESCs) within the skin appears to be an advantage for the adoption of therapies. Use of these cellular types influences the proliferation and migration of fibroblasts and keratinocytes, as well as angiogenesis [60].

In mesenchymal stem cell (**MSC**) therapy, the techniques are based on the use of scaffolds seeded from MSCs and biomaterials such as collagen or cellulose. MSCs stimulate granulation tissue formation, angiogenesis, and future vascularization. Endothelial cells are recruited due to the release of various factors such as FGF or ANG1. The action of MSCs is essentially the modification of the production of TNFalpha and the reduction of the function of natural killer (NK) cells in the inflammatory phase, varying the activity of IFNgamma. If MSCs are used during the last phase of healing, particularly during scar formation, there is a lowering of the TGFbeta1/TGFbeta3 ratio, an increase in IL10, as well as a reduction in IL6 and IL8 [61].

Recently Kua et al. [62] investigated the use of **human umbilical cord cells** for the treatment of skin wounds.

The literature includes various examples describing the action of **drugs** on wound healing. These include anticoagulants, antimicrobials (various classes of antibiotics), anti-angiogenesis agents (e.g., bevacizumab, aflibercept), antineoplastic drugs, antirheumatoid drugs), nicotine, steroids, and vasoconstrictors. Among the various drugs used, steroids and non-steroidal anti-inflammatory drugs (NSAIDs) also deserve attention [63]. Among the new drugs, Exendin 4 (Exe4) is considered an important candidate. Exe4 is a naturally occurring peptide that has 53% similarity with Glucagon-Like Peptide-1 (GLP1), an intestinal insulinotropic peptide that is a member of the incretin hormone family. GLP1 exerts an important postprandial insulinotropic effect, being responsible for about 60% of postprandial insulin secretion. Experimental evidence suggests a possible role for Exe4 in promoting tissue regeneration [64,65].

Extending the use of phototherapy to wound healing is now becoming increasingly important. Lasers are very important in the treatment of assisted healing, most of which use fractional ablative lasers such as CO_2_ or erbium, but non-ablative fractional lasers or vascular lasers are also commonly used [57].

The combination of harmless light in the protoporphyrin absorption spectrum with non-toxic photosensitizing dyes is used in photodynamic therapy (PDT) [48,66].

The role of PDT in CWs has been explored; the main advantage of the technique being the possibility of reducing all types of microorganisms by inducing ROS and without inducing resistance to conventional antibiotics. Furthermore, PDT-induced tissue regeneration and the decrease in metalloproteinase activity and the regeneration of collagen need to be considered [49,50]. However, the use of PDT for assisted WH is not yet habitual in clinical practice as there are few studies published, and multiple repeated sessions are needed with the actual available lights and photosensitizers [66].

Low-level laser light therapy (LLLT) induces cellular modification (photobiomodulation) leading to beneficial clinical effects. LLLT can be applied with low fluences or laser light, but currently, to simplify the use of the technique, light-emitting diodes (LEDs) have been promoted. The action of LLLT is linked to the action of cytochrome c which, by absorbing photons, induces an increase in ATP production and an increase in ROS and transcription factors [67]. However, the role of LLLT remains controversial.

Light-emitting diodes (LEDs) are revolutionizing the entire lighting industry due to their ease of use. However various technical differences distinguish LEDs from LLLTs [68,69,70,71]. The effects of the use of LEDs tend to decrease the inflammatory state of the lesion in progress [68,69,70] and allow for a targeted modulation of the biofilm [65].

**Electrical** stimulation is one of the more promising possible adjuvant therapies, regarding which deeper studies have been published. Some clinical trials have demonstrated the utility of electric field stimulation in CWs such as pressure ulcers and leg ulcers. This technique promotes the proliferation phase and regeneration mainly through the TGFbeta regulation [72]. Ultrasound, delivered in different ways to the target tissue, has also been used as a promising treatment [73,74].

Among the new recent techniques, Bianchi et al. [75] have proposed the use of **nano-fibrous scaffolds**, which have anti-inflammatory properties. The nanofibers come from industrial procedures (electrospinning), using substances such as polybutyl cyanoacrylate (PBCA) combined with copper oxide nanoparticles and casein phosphopeptides (CPP).

Since the use of conventional antibiotics induces antibiotic resistance, Lin et al. [76] developed metal–organic frameworks (**MOFs**), which are complex chemical structures in which copper coordinates with organic ligands.

In contrast, novel isoxazole derivatives or graphene oxide compounds activated by light-emitting diodes were proposed by Bachor et al. [77] and Di Lodovico et al. [78]. These compounds shown anti-microbial activity toward *Staphylococcus aureus* and *Pseudomonas aeruginosa*.

As the use of exosomes as a therapy for wound healing is now clear, Sousa et al. [79], aware of the fact that this therapy has not yet been added to the common treatments, analyzed the substantial efforts that remain necessary before proposing applicable therapeutic approaches using these extracellular vesicles that are produced in the endosomal compartment of most eukaryotic cells.

Finally, considering that in vivo drug studies are limited for many reasons, including ethical ones, Cialdai et al. [80] have developed a new culture system based on enriched culture media and a mechanical support capable of modeling the physiological mechanical tension of the tissue and possibly monitoring its changes during the wound-healing process.

## 9. Future Perspective and Current Limitations

Everything presented in this review may have meaning if we consider that, although it has been known in practice for a very long time, wound healing is a process that has yet to be fully understood despite the scientific literature indicating wound healing as a process of extreme importance that undoubtedly has weight in various branches of medicine.

Therefore, future research should thoroughly investigate the various genetic, biochemical, and physiological questions posed by wound-healing issues about the effect of therapies on these processes. In particular, in vitro or in vivo investigations are needed to understand the cellular and molecular events typical of the processes considered. Of course, of particular interest are the factors involved in local inflammation, perfusion, and hypoxia, as well as cellular interactions. Indeed, therapy based on the use of cells, engineered tissues, or synthetic materials is based on the knowledge of cellular and molecular biology that is applied to the mechanisms involved in wound healing. The integrated knowledge of such therapeutic approaches to promote wound closure may be relevant to the management of chronic wounds. All this is necessary for a physician to integrate the evidence accumulated in the meantime to obtain a complete understanding of the complex wound-healing mechanisms [3,81].

To give an idea of the complexity of the objectives still to be considered, even today, the finest cellular mechanisms are still to be resolved, and the aim of numerous authors is still the identification a cellular reference point, such as fibroblasts or effector cells, while ignoring the fact that knowledge of the skin microenvironment is the key to fully understanding this process [3].

## Figures and Tables

**Figure 2 biomedicines-11-02526-f002:**
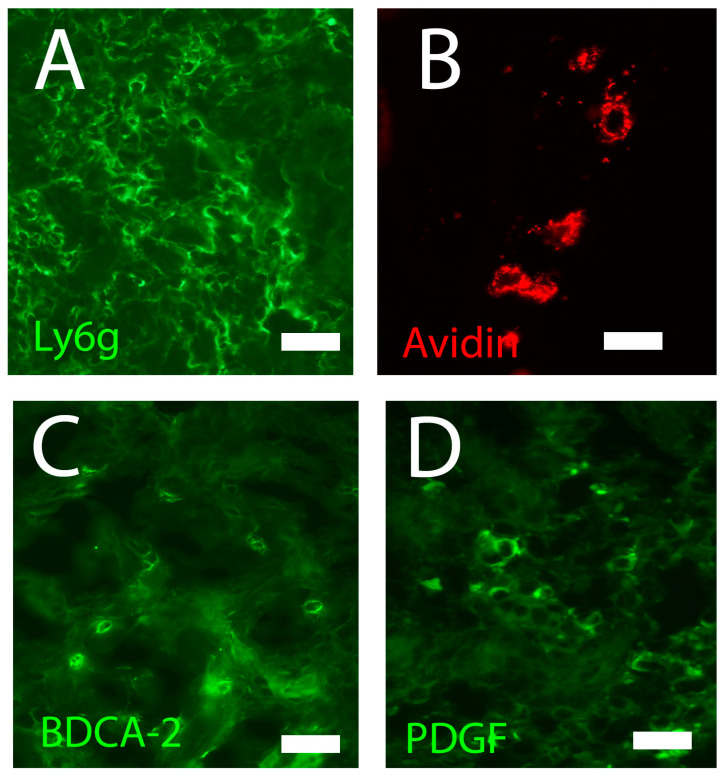
The main cell types that are involved in the composition of the cellular infiltrate during wound healing: (**A**) granulocytes stained with Ly6G (**B**); mast cells stained with avidin; (**C**) plasmacytoid dendritic cells stained with CD303 (BDCA2); (**D**) pericytes stained with PDGF. Fluorescence microscopy; scale bar = 10 microns (courtesy of co-author, Stefano Bacci).

**Figure 3 biomedicines-11-02526-f003:**
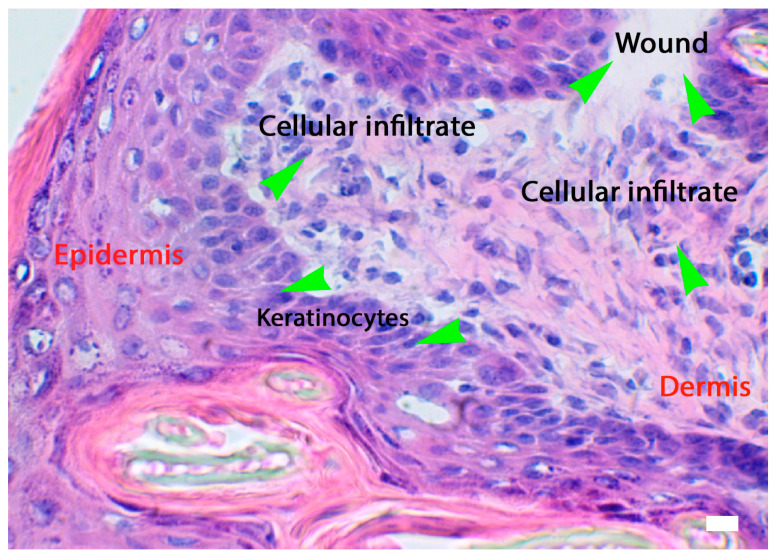
The activation of the skin microenvironment in conditions of injury such as a wound. The responses of keratinocytes and cellular infiltrate (see green arrows for respective locations). HE staining and light microscopy; scale bar = 10 microns (courtesy of co-author, Stefano Bacci).

**Figure 4 biomedicines-11-02526-f004:**
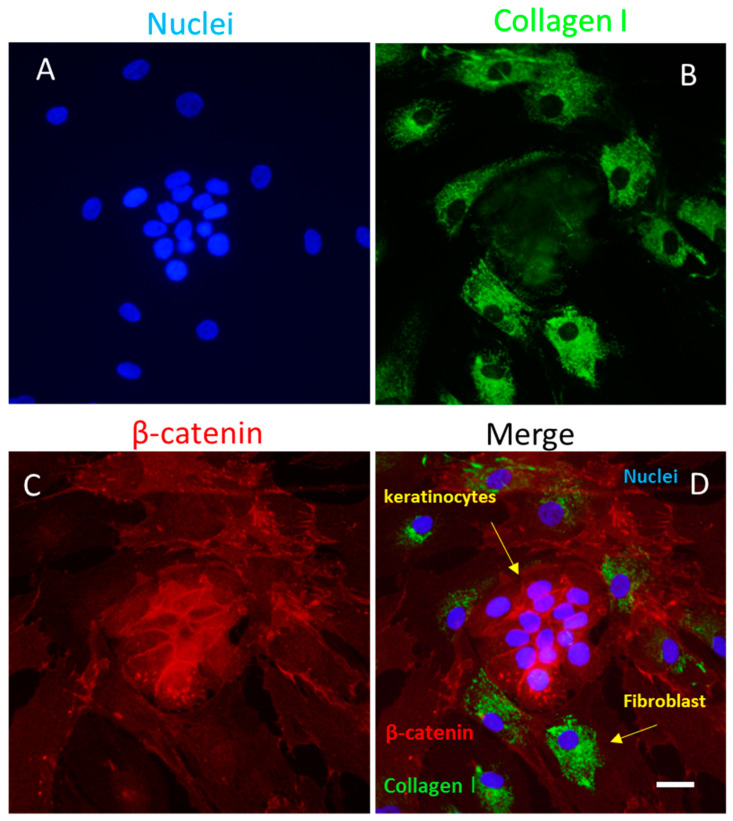
The co-culture of keratinocytes and fibroblasts stained for β-catenin (Alexa red) and collagen type I (Alexa green). (**A**) DAPI-stained nuclei; (**B**) Fibroblasts labeled for collagen type I with Alexa green; (**C**) Keratinocytes labeled for β-catenin with Alexa red; (**D**) merged image. Confocal microscopy; scale bar = 100 μm (courtesy of co-author, Maria Luisa Hernández-Bule).

## Data Availability

Not applicable.

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
