# Peer review of "Cellular and Molecular Processes in Wound Healing"

_biomedicines, 2023, doi:10.3390/biomedicines11092526_

Round 1
Reviewer 1 Report
1. In this review, the authors focus on the more recent advances of the cells and processes in wound Healing. This topic is too big to discuss in detail. I suggest to highlight this review in part of the wound healing process, such as chronic wound healing or others.
2. The topic is more recent advances. However, I didn’t see any recent or novel findings in the paper. Most of discussion is old story.
3. In line 466 and line 470, the full names of ESCs and MSC should have full names in the first appearance.
4. This is a review paper. The authors should provide several schematic cartoon figures to address the cellular or molecular mechanism of wound healing.
5. The authors should focus on the discussion of new growth factors and cytokines involved in wound healing. There is nothing new in this review.
Moderate editing of English language required
Author Response
Review Report Number 1
The authors would like to thank the Referee n. 1 for her/ his efforts and his interesting and constructive comments.
Q1. In this review, the authors focus on the more recent advances of the cells and processes in wound Healing. This topic is too big to discuss in detail. I suggest to highlight this review in part of the wound healing process, such as chronic wound healing or others.
A1. The authors thank the Ref. 1 for the suggestion. To follow the indications we have eliminate two paragraphs that were not appropriate to the goal of the review
Q2. The topic is more recent advances. However, I didn’t see any recent or novel findings in the paper. Most of discussion is old story.
A2. To try to respond to the referee's suggestions we have added, as regards molecular biology, a paragraph dedicated to the action of cytokines and growth factors plus other substances (see Chapter 5 line 292). To this was added another dedicated to the genetic of wound healing (see Chapter 6 line 364). By virtue of this, table 1, which summarizes the most significant data of these paragraphs (including also the secretion of cytokines, growth factor and other substances by the cells involved in the wound healing process), has been modified. Finally, among the various malfunctions of wound healing, space has been dedicated to diabetic wounds (see Paragraph 7.2 line 407). Every paragraph dedicated to these topics is updated.
Q3. In line 466 and line 470, the full names of ESCs and MSC should have full names in the first appearance.
A3 The full names of ESCs and MSC have been added.
Q4. This is a review paper. The authors should provide several schematic cartoon figures to address the cellular or molecular mechanism of wound healing.
A4 The review contains now 2 tables, 1 cartoon and four pictures. The authors are of the opinion that these integrations are able to schematize the various significant steps of the review.
Q5. The authors should focus on the discussion of new growth factors and cytokines involved in wound healing. There is nothing new in this review.
A5. To try to respond to the referee's suggestions we have added, as regards molecular biology, a paragraph dedicated to the action of cytokines and growth factors plus other substances (see Chapter 5 line 292). To this was added another dedicated to the genetics of wound healing (see Chapter 6 line 364). By virtue of this, table 1, which summarizes the most significant data of these paragraphs, has been modified. Finally, among the various malfunctions of wound healing, space has been dedicated to diabetic wounds (see Paragraph 7.2 line 407). The paragraphs dedicated to these topics are updated.

Reviewer 2 Report
The work entitled "Cells and Processes in Wound Healing: more recent advances" aimed to summarize recent knowledge on the main processes that occur during wound healing, including the responses of various cell types and the molecular mechanisms involved. Attention is dedicated to new therapies often involved in the resolution of wounds, whether acute or chronic. The article is generally very basic and would greatly benefit to discuss in more details molecular bases of wound healing. A special chapter should be devoted to chronic diabetic wounds and wound healing in conditions of hyperglycaemia.
The title does not reflect the purpose of the paper, there is no much regarding recent advances and therefore this part should be revised and enriched with an additional chapter, together with future perspectives as well.
3 figures are mentioned and their titles given but just 1 is present in the manuscript, kindly check for errors.
Minor language and style polishing needed. Technical errors detected.
Author Response
Review Report Number 2
The authors would like to thank the Referee n. 2 for her/ his efforts and his interesting and constructive comments.
Q1. The work entitled "Cells and Processes in Wound Healing: more recent advances" aimed to summarize recent knowledge on the main processes that occur during wound healing, including the responses of various cell types and the molecular mechanisms involved. Attention is dedicated to new therapies often involved in the resolution of wounds, whether acute or chronic.
A1. The Ref. 2 understood very well the meaning of this review. Compliments from the authors go to her/him.
Q2. The article is generally very basic and would greatly benefit to discuss in more details molecular bases of wound healing.
A2. To try to respond to the referee's suggestions we have added, as regards molecular biology, a paragraph dedicated to the action of cytokines and growth factors plus other substances (see Chapter 5 line 292). To this was added another dedicated to the genetics of wound healing (see Chapter 6 line 364). By virtue of this, table 1, which summarizes the most significant data of these paragraphs, has been modified. Finally, among the various malfunctions of wound healing, space has been dedicated to diabetic wounds (see Paragraph 7.2 line 407). The paragraphs dedicated to these topics are updated.
Q3. A special chapter should be devoted to chronic diabetic wounds and wound healing in conditions of hyperglycaemia.
A3. This chapter was added at the review (see chapter 8 line 568).
Q4. The title does not reflect the purpose of the paper, there is no much regarding recent advances and therefore this part should be revised and enriched with an additional chapter, together with future perspectives as well.
A4. The title was changed. As already written (see A2) we have added, as regards molecular biology, a paragraph dedicated to the action of cytokines and growth factors plus other substances (see Chapter 5 line 292). To this was added another dedicated to the genetics of wound healing (see Chapter 6 line 364). By virtue of this, table 1, which summarizes the most significant data of these paragraphs, has been modified. Finally, among the various malfunctions of wound healing, space has been dedicated to diabetic wounds (see Paragraph 7.2 line 407). The paragraphs dedicated to these topics are updated. The paragraph dedicated to future perspective and current limitation is in the review.
Q5. Figures are mentioned and their titles given but just 1 is present in the manuscript, kindly check for errors.
A5. Many thanks! Thank you! Now the figures have been placed in the manuscript in the right position.

Round 2
Reviewer 1 Report
1. The cartoon picture below Table 1 is weird. It should be indicated as Figure 1 or others.
2. Why order of Table 1 in the text (line 367) is behind Table 2 (line 122)? I suggest insertion of the word "Table 1" somewhere in the section of Major events in wound healing.
3. The word "as above" in Table 2 is inappropriate. Please revise it.
4. In line 392, Gene Expression in Wound Healing, the authors describe various genes coding for different signaling molecules. However, no specific genes are described in the text.
5. In line 445, Fifteen percent of patients with type 2 diabetes mellitus (T2DM) develop localized 445 ulcers on the lower limbs. Authors should cite references.
6. The full name of LED, MOF should appear in the text.
Author Response
Review Report Number 1
The authors would like to thank the Referee n. 1 for her/ his efforts and his interesting and constructive comments. Every revision of the manuscript has been highlighted (this time in grey)
Q1. The cartoon picture below Table 1 is weird. It should be indicated as Figure 1 or others
A1. We have added, as suggested by the Ref 1 the diction Fig. 1 (see page 4, lines 101-109) and changed the order of other figs in the text.
Q2. Why order of Table 1 in the text (line 367) is behind Table 2 (line 122)? I suggest insertion of the word "Table 1" somewhere in the section of Major events in wound healing.
A2. We have added the diction Table 1 ( see page 9 line 99 and page 10 line 414). Many thanks!
Q3. The word "as above" in Table 1 is inappropriate. Please revise it
A3. We have modified the Table 1. Many thanks!
Q4. In line 392, gene Expression in Wound Healing, the authors describe various genes coding for different signaling molecules. However, no specific genes are described in the text.
A4. The initial chapter (call now genetic activation in wound healing) is now integrated with some paragraphs keeping in mind the interesting observation of Ref. 1 (see pages 15-16: lines 437-495). Therefore, the list of references is now updated.
Q5. In line 445, fifteen percent of patients with type 2 diabetes mellitus (T2DM) develop localized 445 ulcers on the lower limbs. Authors should cite references.
A5. Many thanks for your suggestion. We have added the appropriate references (see page 17, line 539)
Q6. The full name of LED, MOF should appear in the text.
A6. Many thanks for your suggestion. We have added the appropriate full names (see page 19, line 653; see page 20, line 674). We have however checked all the abbreviations and put in a table at the end of the text.

Reviewer 2 Report
The authors made improvements to the manuscript but it still remains basic knowledge. I would suggest implementing more detailed molecular facts and future perspectives for scientific investigations as well.
Abbreviations should be explained when they are used for the first time. Under table 1 there is a figure. Kindly mark it as "figure", not as a part of the table.
Kindly read the abstract once again. Kindly use standard scientific language and methodology in writing abstracts. Besides very unusual sentences, it contains parts/fragments of deleted sentences (e.g. "to define"):
Awareness of these dynamics is important for the 13 various professional figures who are confronted with these kinds of problem daily. However, the 14 study of the healing mechanisms has yet to be defined in detail, in fact there are alterations of var iously coordinated events which lead to a delayed resolution or, as in the case of keloids, to pathological states consisting in the excessive formation of scars with consequences yet to be seen. to define.
English needs polishing by means of style and
Author Response
Review Report Number 2
The authors would like to thank the Referee n. 2 for her/ his efforts and his interesting and constructive comments. Every revision of the manuscript has been highlighted (this time in grey)
Q1. The authors made improvements to the manuscript, but it still remains basic knowledge. I would suggest implementing more detailed molecular facts and future perspectives for scientific investigations as well.
A1. To try to respond to the referee's suggestions regards molecular biology we have:
a) modified (adding other news) the paragraph dedicated to the action of cytokines and growth factors plus other substances (see paragraph 5.1 lines 363-412).
b) modified (adding other news) the paragraph dedicated to the genetic activation in wound healing (see paragraph 5.2 lines 437-495).
c) To these points (a,b) is added table 1 (see page 10 line 414) which represents a summary of the main substances secreted by the cell types involved in wound healing, a topic widely discussed (see pages 4-9, lines 110-362).
About the future perspective for scientific investigation the Chapter 9 ( lines 689-711) has been modified adding other reflections).
Q2. Abbreviations should be explained when they are used for the first time.
A2. Considering the suggestions of the Ref. 2 we have checked all the abbreviations and again the list of them.
Q3. Under table 1 there is a figure. Kindly mark it as "figure", not as a part of the table.
A3. We have added, as suggested by the Ref 2 the diction Fig. 1 (see page 4, lines 101-109) and changed the order of other figs in the text.
Q4. Kindly read the abstract once again. Kindly use standard scientific language and methodology in writing abstracts. Besides very unusual sentences, it contains parts/fragments of deleted sentences (e.g. "to define"): Awareness of these dynamics is important for the 13 various professional figures who are confronted with these kinds of problem daily. However, the 14 study of the healing mechanisms has yet to be defined in detail, in fact there are alterations of var iously coordinated events which lead to a delayed resolution or, as in the case of keloids, to pathological states consisting in the excessive formation of scars with consequences yet to be seen. to define.
A4. The referee is completely right, and we apologize. We have re-read the abstract again and made some changes

Round 3
Reviewer 1 Report
No
Author Response
Review Report Number 1
The authors would like to thank the Referee n. 1 for her/ his efforts and his interesting and constructive comments. Every revision of the manuscript has been highlighted (this time in green)
Q1. Comments and Suggestions for Authors: No
A1. Many thanks!

Reviewer 2 Report
The authors made improvements to the text. English language should be professionally revised.
Abbreviations are NOT consistent in the manuscript, e.g. line 169 TNF-@, IL-1B, IL-6 and 6 lines below, line 175: IL1B for example. ALL abbreviations should be checked and corrected. Many of them are not used in an appropriate way. Differences were also found in the new table.
English language should be professionally revised.
Author Response
Review Report Number 2
The authors would like to thank the Referee n. 2 for her/ his efforts and his interesting and constructive comments. Every revision of the manuscript has been highlighted (this time in green)
Q1. The authors made improvements to the text. English language should be professionally revised.
A1 The manuscript has been edited for English style (See attached certificate in the present letter). The authors propose however a new title for the manuscript. This is due to the need on the part of the authors to maintain consistency with what has been improved thanks to the suggestions of both the Editor and the reviewers. This change happened later the editing for English style.
Q2: Abbreviations are NOT consistent in the manuscript, e.g. line 169 TNF-@, IL-1B, IL-6 and 6 lines below, line 175: IL1B for example. ALL abbreviations should be checked and corrected. Many of them are not used in an appropriate way. Differences were also found in the new table.
A2. Considering the suggestions of the Ref. 2 we have checked all the abbreviations and again the list of them.
